# Microwave-Assisted Synthesis of MoS_2_/BiVO_4_ Heterojunction for Photocatalytic Degradation of Tetracycline Hydrochloride

**DOI:** 10.3390/nano13091522

**Published:** 2023-04-30

**Authors:** Cixin Cheng, Qin Shi, Weiwei Zhu, Yuheng Zhang, Wanyi Su, Zizheng Lu, Jun Yan, Kao Chen, Qi Wang, Junshan Li

**Affiliations:** 1Guangxi Colleges and Universities Key Laboratory of Environmental-Friendly Materials and New Technology for Carbon Neutralization, Guangxi Key Laboratory of Advanced Structural Materials and Carbon Neutralization, School of Materials and Environment, Guangxi Minzu University, Nanning 530105, China; 2Guangxi Research Institute of Chemical Industry Co., Ltd., Nanning 530006, China; 3Faculty of Material Science and Engineering, Kunming University of Science and Technology, Kunming 650093, China; 4School of Chemistry and Chemical Engineering, Guangxi Minzu University, Nanning 530006, China; 5Key Laboratory of Advanced Materials of Yunnan Province, Kunming 650093, China; 6Institute for Advanced Study, Chengdu University, Chengdu 610106, China

**Keywords:** MoS_2_/BiVO_4_ heterojunction, microwave-assisted synthesis, degradation, tetracycline hydrochloride, photocatalysis

## Abstract

Compared with traditional hydrothermal synthesis, microwave-assisted synthesis has the advantages of being faster and more energy efficient. In this work, the MoS_2_/BiVO_4_ heterojunction photocatalyst was synthesized by the microwave-assisted hydrothermal method within 30 min. The morphology, structure and chemical composition were characterized by X-ray diffraction (XRD), Raman, X-ray photoelectron spectroscopy (XPS), scanning electron microscope (SEM), and high-resolution transmission electron microscopy (HRTEM). The results of characterizations demonstrated that the synthesized MoS_2_/BiVO_4_ heterojunction was a spherical structure with dimensions in the nanorange. In addition, the photocatalytic activity of the samples was investigated by degrading tetracycline hydrochloride (TC) under visible light irradiation. Results indicated that the MoS_2_/BiVO_4_ heterojunction significantly improved the photocatalytic performance compared with BiVO_4_ and MoS_2_, in which the degradation rate of TC (5 mg L^−1^) by compound where the mass ratio of MoS_2_/BiVO_4_ was 5 wt% (MB5) was 93.7% in 90 min, which was 2.36 times of BiVO_4_. The active species capture experiments indicated that •OH, •O_2_^−^ and h^+^ active species play a major role in the degradation of TC. The degradation mechanism and pathway of the photocatalysts were proposed through the analysis of the band structure and element valence state. Therefore, microwave technology provided a quick and efficient way to prepare MoS_2_/BiVO_4_ heterojunction photocatalytic efficiently.

## 1. Introduction

The 2022 Global Environmental Performance Index (EPI) issued in June 2022 pointed out that two billion people in the world could not get clean drinking water due to the shortage of water resources and water pollution [1]. Wastewater treatment is a major global problem. Antibiotics from livestock raising [2], sewage treatment plants [3] and other channels are rampant in the water environment [4]. Tetracycline hydrochloride (TC) is a major broad-spectrum antibiotic, cannot be removed through spontaneous degradation because of its high structural stability, and its existence can bring potential risks to plants and human health [5]. Many methods have been used to remove TC from water, including photocatalysis [6], adsorption [7], biodegradation [8], and efficient activation of peroxydisulfate (PS) [9]. In comparison, photocatalytic technology is one of the practical techniques for the degradation of contaminants in water, due to the advantage of its high-efficiency and no secondary pollution [10].

Since BiVO_4_ was applied as a photocatalyst [11], it has attracted significant attention in the field of photocatalysis. BiVO_4_ as an n-type semiconductor, is considered to be a promising visible-light-driven photocatalyst with a low band gap (E_g_ = 2.4 eV), sensitive visible light response, strong chemical stability, and nontoxicity [12]. However, the photocatalytic performance of BiVO_4_ has limited by the high photogenerated carriers recombination rate and weak surface adsorption ability [13]. As previous reports [14,15] suggest, building heterojunctions is an effective way to solve the above problems of BiVO_4_. 

Creating heterostructures by coupling more than two semiconductors with matched band positions is a way to significantly improve the photocatalytic capability of single photocatalysts. For instance, Guo et al. [16] prepared the BiVO_4_/Bi_0.6_Y_0.4_VO_4_ heterojunction by an in-situ pressure method following by hydrothermal, which significantly enhanced the overall photocatalytic water splitting activity. Jin et al. [17] used a self-assembly method to construct a La-BiVO_4_/CN step-scheme heterojunction photocatalyst and applied the photocatalyst to degrade TC, which showed enhancing photocatalytic activity. Hu et al. [18] fabricated a g-C_3_N_4_/BiVO_4_ heterojunction photocatalyst by hydrothermal synthesis, where the degradation rate of benzyl-paraben by g-C_3_N_4_/BiVO_4_ was 1.42 times higher than that of BiVO_4_ under direct natural sunlight. Therefore, the construction of heterostructures between BiVO_4_ and semiconductors with a narrow band gap could effectively improve the photocatalytic ability of BiVO_4_.

As a 2D-layered and p-type semiconductor, molybdenum disulfide (MoS_2_) has a high surface area and excellent adsorptive properties [19]. MoS_2_ also has a narrow and adjustable band gap between 1.2–1.9 eV benefited by different layer structures [20], thus having a nice visible light absorption capacity. To date, the MoS_2_/BiVO_4_ heterojunction has been developed for the photocatalytic degradation of methylene blue [21], tetracycline [22], bisphenol A [23] and other pollutants [24,25] with excellent degradation results. According to some previous reports, the techniques used for constructing MoS_2_/BiVO_4_ heterojunction materials such as hydrothermal synthesis [21], electrospinning [24], and the ultrasonic agitation [26] have the disadvantages of being time consuming and technically complex. Microwave synthesis is a promising synthesis method. Microwave radiation can improve the reaction yields by penetrating materials and coupling directly with ions [27]. In addition, microwave synthesis has the advantages of a rapid temperature rise, uniform heat transfer, high controllability, and quick reaction rate [28,29]. The microwave synthesis has been used for the preparation of heterojunction materials, such as BiOI/BiF_3_ [30], CdS/BiOBr [31], La(OH)_3_/BiOCl [32], BiVO_4_/WO_3_ [33], and so on. Sriram et al. [34] used a hydrothermal-microwave synthesis to prepare a MoS_2_/BiVO_4_ modified electrode, and then measure 3-nitro-L-tyrosine in a biological media with this electrode without any pretreatment. So far, we have not seen any reports on the degradation of environmental pollutants by microwave synthesis of MoS_2_/BiVO_4_ heterojunction. We used a microwave-assisted hydrothermal method which was similar to Sriram to prepare MoS_2_/BiVO_4_ photocatalyst for photocatalytic degradation of pollutants in water.

The preparation of MoS_2_/BiVO_4_ heterojunction by the hydrothermal method usually takes several hours. In this paper, a MoS_2_/BiVO_4_ heterojunction was efficiently constructed via a microwave-assisted synthesis within 30 min. Additionally, the MoS_2_/BiVO_4_ photocatalysts were applied to the degradation of TC under visible light. The morphology, structure and element composition of the samples were determined by a series of characterizations. In addition, the free radical quenching experiments were carried out to investigate the reaction mechanism. The MoS_2_/BiVO_4_ heterojunction photocatalyst showed superior adsorption and visible light photocatalytic capability for TC. 

## 2. Experimental

### 2.1. Materials

Thiourea (CH_4_N_2_S), sodium molybdate (Na_2_MoO_4_), bismuth nitrate hydrate (Bi(NO_3_)_3_·5H_2_O), ethylene glycol ((CH_2_OH)_2_), ammonium metavanadate (NH_4_VO_3_), tetracycline hydrochloride (TC, C_22_H_25_ClN_2_O_8_), tert-butyl alcohol (TBA, C_4_H_10_O), ammonium oxalate (AO, (NH_4_)_2_C_2_O_4_), and p-benzoquinone (PBQ, C_6_H_4_O_2_) were obtained from Macklin Biochemical Co., Ltd., (Shanghai, China). The reagents used were all analytically pure and all water used for experiments was ultrapure water.

### 2.2. Synthesis of MoS_2_ Nanosphere

The hydrothermal treatment was used to manufacture the MoS_2_ nanoflower. Na_2_MoO_4_ (0.6096 g) and CH_4_N_2_S (0.9104 g) were dissolved in 60 mL ultrapure water. Subsequently, the mixing solution was transferred into a 100 mL Teflon-lined autoclave and heated under 200 °C for 24 h. The product was cooled to room temperature and washed with ethanol and ultrapure water. The obtained black compound was dried at 60 °C for 12 h. 

### 2.3. Preparation of MoS_2_/BiVO_4_ Heterojunction

Bi(NO_3_)_3_·5H_2_O (3.1920 g) and NH_4_VO_3_ (0.9453 g) were added to 80 mL (CH_2_OH)_2_, then stirred until fully dissolution. Then, a certain mass of MoS_2_ (0.0099, 0.0168, 0.0241 g) was added to the aforementioned solution and ultrasound dispersion. The above mixing solution was transferred into a reaction bottle and reacted at 300 W and 120 °C reacted for 30 min in a microwave reactor (CEM-Discover SP, CEM, Matthews, NC, USA). The product was cooled to room temperature and washed with ethanol and ultrapure water several times. Obtained yellow compound was dried at 60 °C for 12 h. Composites with different loading amounts of MoS_2_ were prepared by adding different quality of MoS_2_, which were denoted as MB3, MB5 and MB7, respectively (“3” meant the mass ratio of MoS_2_/BiVO_4_ was 3 wt%).

### 2.4. Characterization and Analysis Methods

The surface morphology of the prepared materials was characterized by a scanning electron microscope (SEM, SUPRA 55 Sapphire, Zeiss, Oberkochen, Germany). X-ray diffractometer (XRD, D8 advance, Bruker, Karlsruhe, Germany) was employed to test physical phase and crystalline size of the materials at the 2*θ* range of 10–80° at a step size of 0.02° and a scan rate of 6°/min (Cu Ka radiation, λ = 0.15814 nm). Raman spectroscopy (Raman, inVia Reflex, Renishaw, Gloucestershire, UK) was used to investigate molecular structure of the samples with an emission wavelength of 532 nm, 10% laser power, and the scanning range was 200–1000 cm^−1^. Using FT-IR spectrometer (IRAffinity-1s, Shimadzu, kyoto, Japan) with the scan range from 2000 to 400 cm^−1^ and the resolution of 4 cm^−1^ to test the chemical bond type and structure of materials. A UV-vis spectroscopy (UV-2700, Shimadzu, Japan) with a range of 200–800 nm was used to examine UV-visible absorption spectra of catalysts. The specific surface area of the samples was studied by Brunner-Emmet-Teller (BET) measurements (ASAP2460, Micromeritics, Norcross, GA, USA). The elemental composition and changes of the sample surface were investigated by X-ray photoelectron spectroscopy (XPS, Axis Ultra DLD Kratos AXIS SUPRA, Shimadzu, Japan).

### 2.5. Photocatalytic Activity Tests

The visible-light source was generated by using a Xenon lamp (300 W, China Education AU-Light, Beijing, China) with a 420 nm cut-off filter to remove light of λ < 420 nm. The photocatalytic degradation of TC was used to test photocatalytic performance of the prepared samples. A double-layer beaker with a water circulation cooling system was used as the container. Specifically, 50 mg of photocatalyst was added to 100 mL of TC solution (5 mg L^−1^) was stirred in dark for 30 min to achieve the adsorption-desorption equilibrium. And then turned on the light illuminated 90 min for photocatalytic reaction. The absorption at 356 nm was used to analyze the TC concentration [35]. The active species capture experiments were performed to explore the major active species in the MB5 photocatalyst and the mechanism of photocatalytic degradation of TC. The active species capture agents were TBA (1.8 mL), AO (0.3 mmol), and PBQ (0.3 mmol), which were used to capture •OH, h^+^ and •O_2_^−^, respectively.

### 2.6. PEC Spectra Measurements

Electrochemical tests were performed in a standard three-electrode system using a CHI 760E electrochemical workstation (CH Instruments Ins., Shanghai, China). A Pt electrode was used as a counter electrode, and an Ag/AgCl electrode was used as a reference electrode. A total of 50 mg of photocatalyst was dispersed in 2 mL of ultrapure water by ultrasound, then dropped 100 μL of the mixed solution on to FTO glass, where the resultant electrode used as a working electrode [21]. In this study, Na_2_SO_4_ (0.1 mol L^−1^) solution was employed as the electrolyte. A Xenon lamp light source (300 W) with a 420 nm cut-off filter to remove light of λ < 420 nm was used as the visible-light source. Nyquist plots tested by scanning the frequency range were 100 mHz–10 kHz with a voltage amplitude of 5 mV. The applied voltage is 1 V vs. Ag/AgCl during current versus time curve (i-t) for testing photocurrent density.

## 3. Results and Discussion

### 3.1. Characterization

The crystal phase of the samples was characterized by XRD. As shown in Figure 1, the diffraction peaks of BiVO_4_, and all MoS_2_/BiVO_4_ heterojunctions had a good corresponding relationship with the monoclinic scheelite BiVO_4_ (JCPDS No. 14-0688) [36]. The diffraction peaks at 2*θ* = 14.0°, 33.0° and 58.3° of MoS_2_ are derived from the contributions of the (002), (100) and (110) crystal planes of hexagonal MoS_2_ (JCPDS No.37-1492) [37]. Remarkably, when the loading amount of MoS_2_ was low, the diffraction peaks of MoS_2_ were absent in the XRD spectra of MB3 and MB5. As the amount of MoS_2_ continued to increase, the diffraction peak located at 33.0° corresponds to the (100) crystal plane of MoS_2_ which appeared in MB7 composites. Moreover, it proved the successful preparation of MoS_2_/BiVO_4_ [38]. The enhancement of the diffraction peak intensity of the (121) crystal plane in MB5 and MB7 was significantly greater than the other diffraction peaks, and this probably attributed to the addition of MoS_2_ affected the crystal structure and the growth direction of BiVO_4_ [39,40] The grain size can be calculated by the Debye-Scherrer equation [41] (Equation (1)),
(1)$$D=\frac{K\lambda }{\mathrm{Bcos}\theta }$$
where *D* is the average grain size, *K* = 0.89, *λ* = 1.5406 nm, *B* is the full width at half maximum of the diffraction peaks, *θ* is the Bragg’s angle, which in this study is the diffraction angle corresponding to the (121) crystal plane. The average crystallite sizes of BiVO_4_, MB3, MB5 and MB7 were calculated as 6.9, 2.9, 5.5, 17.2 nm, respectively. The addition of MoS_2_ influenced the particle size of MoS_2_/BiVO_4_ composites, where the smallest size was MB3.

Raman spectroscopy analysis (Figure 2a) and FT-IR spectra (Figure 2b) could be helpful to further validate the structure of samples. As shown in Figure 2a, BiVO_4_ and MoS_2_/BiVO_4_ composites have a strong peak at 820 cm^−1^ which is caused by V-O bond tensile vibration, and the peak at 340 cm^−1^ is the asymmetric and symmetric bending vibration of VO_4_^3−^ [42]. Further, the characteristic peaks of MoS_2_ at 378 and 406 cm^−1^ correspond to the E^1^_2g_ and A_1g_ vibrational modes [43]. In Raman spectra, red shifts in the MoS_2_/BiVO_4_ spectral band can be clearly seen, the peak at 815 cm^−1^ has an increase of the intensity, and along with the stretching vibration of the V-O bond and the bending vibration of VO_4_^3−^ become irregular, which can be attributed to the interaction between MoS_2_ and BiVO_4_. According to the available literature [44,45], such a red shift and increase in peak intensity of MB5 were due to successful preparation of heterojunction. FT-IR spectra of MoS_2_, BiVO_4_ and MB5 were shown in Figure 2b. For BiVO_4_, the peak at 416 cm^−1^ corresponds to the chemical stretching of Bi-O. The absorption band observed at 745 cm^−1^ was attributed to VO_4_^3−^ symmetric and tensile vibration peak [46]. For MoS_2_, the peaks at 420, and 1401 cm^−1^ were allocated to S-S and S-Mo-S bond vibration and tensile vibration [47]. For MoS_2_/BiVO_4_ composites, all single-phase vibrations corresponded well.

SEM was employed to observe the micro morphology of the photocatalyst (Figure 3). Figure 3a indicated that MoS_2_ is a nanosphere structure with an average diameter of about 4 μm. Figure 3b showed BiVO_4_ as a nanosphere structure with the size of about 0.5 um. The plots of MoS_2_/BiVO_4_ with different ratios were placed in Figure 3c–e, when MoS_2_ nanospheres were added to the microwave reactor, they were separated into nano-microspheres of smaller size by microwave radiation. It can be observed that the surface morphology of the MoS_2_/BiVO_4_ heterojunction with a different mass ratio of MoS_2_ and BiVO_4_ is different. First, MB3 was similar to BiVO_4_ nanospheres, and the surface of MB3 that MoS_2_ was completely covered by BiVO_4_ (Figure 3c). In the second place, some irregular depressions could be observed on the surface of MB5 (Figure 3d). Finally, with MB7 it could be seen that there were many MoS_2_ exposed on the surface, which was due to the fact that as the added MoS_2_ increases MoS_2,_ it was not completely covered by BiVO_4_ (Figure 3e). The results showed that the growth process and the structure are similar to MoS_2_/BiVO_4_ heterojunction prepared by Peng et al. [25]. During the reaction, with MoS_2_ as a substrate, BiVO_4_ would nucleate and continue to grow on the surface of MoS_2_ nano-microspheres, eventually covered the entire surface of MoS_2_, and grew into a nanosphere structure.

Further investigation of the microstructure of MoS_2_/BiVO_4_ heterostructures was conducted through HRTEM (Figure 4). The interplanar spacing of 0.31 nm and 0.27 nm were clearly found in Figure 4b, which were correspond to the (121) plane of BiVO_4_ [48] and the (100) plane of MoS_2_ [49], respectively. In addition, the chemical composition and elemental distribution of MB5 were analyzed by SEM-EDS. As shown in Figure 4c–g, the MB5 composites photocatalyst was composed of five elements (Bi, V, O, Mo, and S). The above images could be clearly observed that BiVO_4_ and MoS_2_ formed an interface in close contact further provided evidence for the successful preparation of MoS_2_/BiVO_4_ heterostructures.

With the use of XPS, the surface molecular structure and chemical states of MoS_2_, BiVO_4_, and MB5 were identified. The presence of Bi, V, O, Mo, and S element in the MB5 were revealed by the full scan XPS spectrum (Figure 5a). Figure 5b shows the Mo 3d high resolution XPS spectra. Mo 3d_5/2_ and Mo 3d_3/2_ of MB5 were matched with two peaks at 232.1 eV and 235.2 eV. The energy gap of these two peaks is approximately 3.1 eV, indicating that molybdenum ions are Mo^4+^ in a lower oxidation state [25]. There is no characteristic peak of Mo3d in BiVO_4_, while it is present in MB5, which proves the presence of MoS_2_ in the MoS_2_/BiVO_4_ heterojunction. In Figure 5c, two significant peaks that occurred around 164.3 eV and 159.0 eV were identified to the Bi 4f_5/2_ and Bi 4f_7/2_; the energy gap of 5.3 eV confirmed that the bismuth species occurred as Bi^3+^ in MB5. Vanadium ions in MB5 existed in the shape of V^5+^. Two significant peaks near 524.2 eV and 516.6 eV were attributed to the V 2p split signals (V 2p_1/2_ and V 2p_3/2_) clearly seen in Figure 5e [24]. Likewise, the main peak of O 1s in the high resolution XPS spectra could be divided into two bands, implying two different types of oxygen existed on the surface of MB5 [23]. The XPS results indicate a strong interaction between MoS_2_ and BiVO_4_ and the successful preparation of a heterostructure.

Specific surface area is an important factor affecting the catalytic efficiency of photocatalysts. Figure 6 is the N_2_ adsorption-desorption curve and the pore size distribution map of MoS_2_, BiVO_4_ and MB5 measured by the Brunauer-Emmett-Teller (BET) and Barret-Joyner-Halenda (BJH), respectively. As shown in Figure 6a, all photocatalysts showed type-IV isotherms with H_3_ hysteresis loops [50]. The specific surface areas of 9.79 m^2^ g^−1^, 5.07 m^2^ g^−1^, and 11.02 m^2^ g^−1^ were for MoS_2_, BiVO_4_, and MB5, respectively. The findings indicated that MB5 had the largest specific surface area. This was because in the synthesis process of MB5, where MoS_2_ were broken down into smaller-sized MoS_2_ by microwave energy. Through the microwave reaction, the increased surface area of the MoS_2_ led to an overall increase in the specific surface area of MB5 [51]. The large specific surface area provides more active sites and higher adsorption capacity. In addition, Figure 6b showed the presence of a mesoporous structure, which could serve as a fast transfer path for photogenerated electrons [52].

### 3.2. Light Absorption and Charge Transfer Performance

The optical features of the photocatalysts could be assessed by using UV-visible diffuse reflectance spectroscopy (UV-vis DRS). In Figure 7a, BiVO_4_ had an absorption band at 500 nm. MoS_2_ showed a response to ultraviolet and visible light region, indicating its wide absorption rage [53]. MB5 had an increased absorption capacity in both the ultraviolet and visible light region compared with BiVO_4_, contributing to the higher utilization of solar light. Tauc plot (Figure 7b) showed the band gaps of 1.29 eV, 2.31 eV and 2.25 eV for MoS_2_, BiVO_4_ and MB5, respectively. Compared to BiVO_4_, MB5 had a reduced band gap. According to the above analysis, compared with MoS_2_ and BiVO_4_, electrons in the MB5 were more easily excited from the valence band to the conduction band due to the stronger optical absorption capability and the reduced band gap width. In order to reveal the photogenerated charge divorce process of the heterojunction in the photocatalysis, the conduction band (CB) and valence band (VB) of BiVO_4_ and MoS_2_ were calculated by the following formula [54] (Equations (2) and (3)):(2)$$E_{CB}=X-E^{e}-0.5E_{g}$$
(3)$$E_{VB}=E_{CB}+E_{g}$$
where, *E^e^* is the energy of free electrons at hydrogen standard potential, 4.5 eV. *E_g_* is band gap. *X* is the absolute electronegativity of the semiconductor. The *X* values of BiVO_4_ and MoS_2_ are 6.16 eV and 5.32 eV, which are calculated by the following formula (Equation (4)):(4)$$X={{\left[x(A\right)}^{a}{x(B)}^{b}{x(C)}^{c}]}^{1/(a+b+c)}$$
in which *x*(*Bi*) = 4.69 eV, *x*(*V*) = 3.60 eV, *x*(*O*) = 7.54 eV, *x*(*Mo*) = 3.90 eV, *x*(*S*) = 6.22 eV. *a*, *b*, and *c* are the number of atoms in compound [55].

The calculated CB values of BiVO_4_, MoS_2_ are 0.505 eV and 0.175 eV, respectively. The calculated VB values of BiVO_4_ is 2.815 eV, and MoS_2_ is 1.465 eV. Apparently, MoS_2_ and BiVO_4_ could form a heterojunction because they have a staggered band position.

To further investigate the charge separation of MoS_2_/BiVO_4_ heterojunction, the photocurrent response and electrochemical impedance spectroscopy (EIS) of MoS_2_, BiVO_4_ and MB5 were measured (Figure 8). Figure 8a showed that, MB5 displayed the highest photocurrent response (1.3 μA cm^−2^) under visible light (λ > 420 nm) illumination, which was 3.25 times that of MoS_2_ and 1.85 times that of BiVO_4_. A higher photocurrent response of the MB5 suggests the increased charges separation efficiency, which increased the lifetime of the photogenerated charge and thus improved the photocatalytic rate [56,57]. In the EIS curve, when compared with MoS_2_ and BiVO_4_, it was evident that MB5 had the smallest arc radius (Figure 8b), which was consistent with the results of the photocurrent density test. The minimum arc radius implies that MB5 has the lowest resistance and highest charge transfer efficiency. The results suggested that the separation efficiency and transport performance of photogenerated carriers in heterojunction were hugely enhanced.

### 3.3. Catalytic Capacity of MoS_2_/BiVO_4_

To assess the visible light photocatalytic activity of BiVO_4_, MB3, MB5, and MB7, TC degradation experiments were carried out. Figure 9a indicated that 39.7% of TC could be degraded in 90 min by BiVO_4_. In the same conditions, 93.7% of TC could be degraded by MB5, it is 2.36 times greater than BiVO_4_. Moreover, with the increase of MoS_2_ compounded in MoS_2_/BiVO_4_, the degradation rate of TC by MoS_2_/BiVO_4_ photocatalyst showed a trend of increasing and then decreasing, and MB5 showed the best enhancement of photocatalytic activity. Compared with BiVO_4_, there may be three reasons for the enhanced photocatalytic performance of MB5: firstly, the larger specific surface area signifies an increased adsorption capacity, secondly, the improved utilization of visible light and the narrower band gap boosts the excitation of electrons, and thirdly, the generation of transport channels between MoS_2_ and BiVO_4_ are beneficial to the rapid transfer and separation of photogenerated electron-hole pairs. Figure 9b showed the result of the TC in three cyclic degradation experiments. Due to the loss of photocatalytic materials in the recycling process, the degradation rate decreases with the number of uses. After the catalyst was reused three times, the degradation efficiency remained above 80%. By a pseudo first-order model (lnC_0_/C = kt) [58], the degradation processes were fitted. Figure 9c showed that all fitted curves are nearly linear. As shown in Figure 9d, the kinetic constants k for the synthesized BiVO_4_, MB3, MB5 and MB7 are 0.0015, 0.0031, 0.0215 and 0.0012 min^−1^, respectively. The kinetic constants k of MB5 is much higher than other photocatalysts, which was 14.3 times that of BiVO_4_. The above results illustrated the effectiveness and stability of MoS_2_/BiVO_4_ heterojunction as a photocatalyst. 

### 3.4. Photocatalytic Mechanism Study

A radical capture test was used to explore the photocatalytic mechanism. Under the same experimental conditions, TBA, AO and PBQ were added as radical trapping agents for •OH, h^+^ and •O_2_^−^, respectively [59,60]. The outcomes in Figure 10 revealed that after the addition of the three catchers TBA, AO, and PBQ, respectively, the degradation efficiency reduced to 54%, 13% and 29%. It could be concluded from the test results that the efficient degradation of TC benefited from the synergistic effect of •OH, h^+^, and •O_2_^−^. Among them, h^+^ had two working modes, one was the direct oxidative degradation of TC, and the other was reaction with the water adsorbed on the surface of the material to form •OH, which then oxidizes and degrades TC. Owing to the addition of AO had the most pronounced inhibitory effect on the degradation of TC, which indicated that the primary active species for the oxidative degradation TC was h^+^ in the photocatalytic reaction for MB5.

Hence, based on the relative positions of energy bands, and the above radical catch test results, possible photodegradation mechanisms of MoS_2_/BiVO_4_ photocatalysis is posited (Figure 11). The p-type MoS_2_ and the n-type BiVO_4_ form a p-n heterojunction, and MoS_2_ had higher CB and VB than BiVO_4_ based on the energy band structure. Firstly, because the adsorption property of MB5 was enhanced by the addition of MoS_2_, the large amount of TC was adsorbed on the surface of the photocatalytic materials. Under visible light irradiation, MoS_2_ and BiVO_4_ were excited simultaneously. The photogenerated holes of BiVO_4_ and MoS_2_ degraded TC as h^+^ radicals. And some of the photogenerated holes in the VB of BiVO_4_ migrated to the VB of MoS_2_ and then formed •OH with water or hydroxyl groups which was adsorbed on the surface of the material. MoS_2_ and BiVO_4_ originally had different Fermi levels, and the Fermi levels became the same after the heterojunction was formed, which moved the CB of MoS_2_ up to a higher level [61]. Therefore, in the heterojunction, the e^−^ in the CB of MoS_2_ had a stronger redox ability to react with dissolved oxygen to form •O_2_^−^. The matching energy band position between MoS_2_ and BiVO_4_ allows the generation of fast separation channels for photogenerated carriers at the heterojunction interface, which contributes to the generation of •OH, h^+^ and •O_2_^−^. Moreover, h^+^ played the most dominant role in the degradation of TC. Finally, the three active substances synergize with each other to oxidize and degrade the pollutants which are adsorbed on the surface of the photocatalytic materials into H_2_O and CO_2_.

## 4. Conclusions

Herein, a p-n heterojunction photocatalyst MoS_2_/BiVO_4_ was obtained by microwave-assisted synthesis of BiVO_4_ crystals that were grown on the surface of MoS_2_ nano-microspheres. Compared with BiVO_4_, the heterojunction structure formed by MoS_2_/BiVO_4_ has a stronger adsorption capacity, greater visible light utilization, smaller band gap and higher separation of photogenerated carriers, and these advantages synergistically lead to the enhanced photocatalytic activity of the MoS_2_/BiVO_4_. For the above reasons, MB5 showed better photocatalytic activity under visible light, and the degradation rate of TC reached 93.7% within 90 min, and the first-order kinetic constant of MB5 was 0.0215 min^−1^, which was 14.3 times higher over BiVO_4_. Cyclic degradation experiments demonstrated the reusability and stability of the MoS_2_/BiVO_4_ photocatalyst. This study provides an energy-saving and an efficient, feasible method for the preparation of the MoS_2_/BiVO_4_ heterojunction photocatalyst with excellent photocatalytic performance, and the MoS_2_/BiVO_4_ heterojunction photocatalyst is expected to realize broader applications in the wastewater treatment.

## Figures and Tables

**Figure 1 nanomaterials-13-01522-f001:**
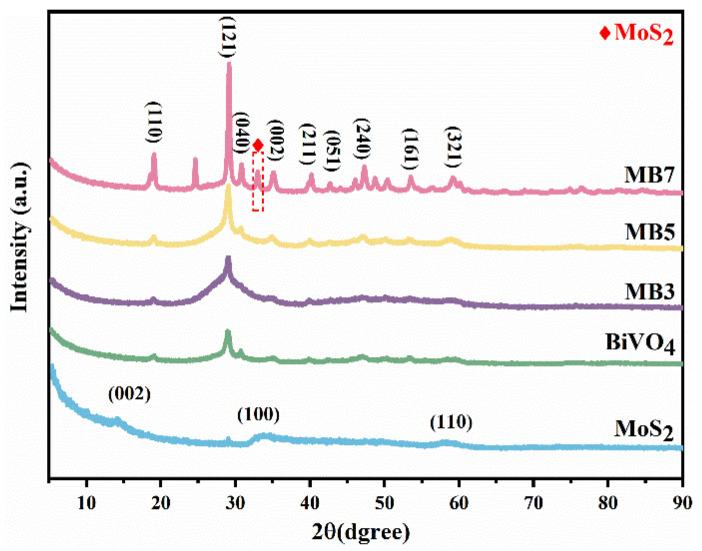
XRD patterns of MoS_2_, BiVO_4_, MB3, MB5 and MB7 (♦ mark shows the presence of MoS_2_).

**Figure 2 nanomaterials-13-01522-f002:**
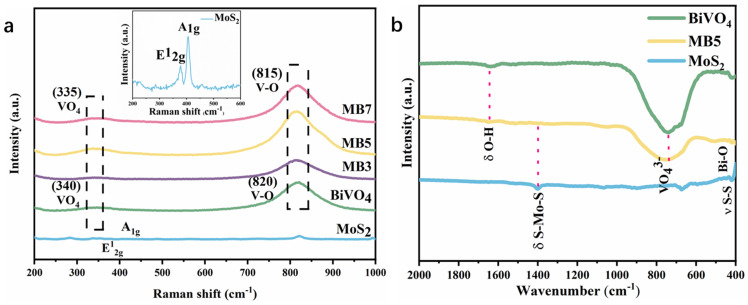
(**a**) Raman spectrum of MoS_2_, BiVO_4_, MB3, MB5 and MB7 and (**b**) FT−IR spectra of MoS_2_, BiVO_4_ and MB5.

**Figure 3 nanomaterials-13-01522-f003:**
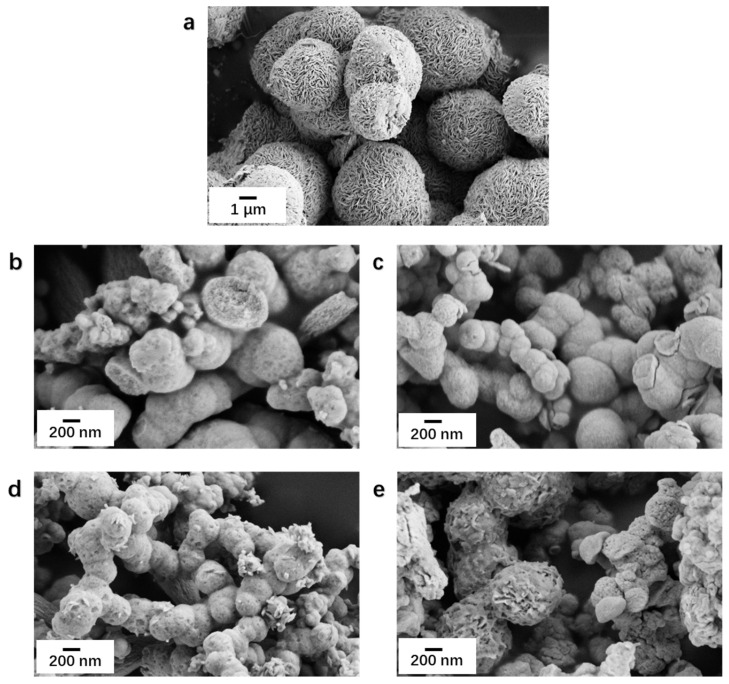
SEM image of (**a**) MoS_2_, (**b**) BiVO_4_, (**c**) MB3, (**d**) MB5, and (**e**) MB7.

**Figure 4 nanomaterials-13-01522-f004:**
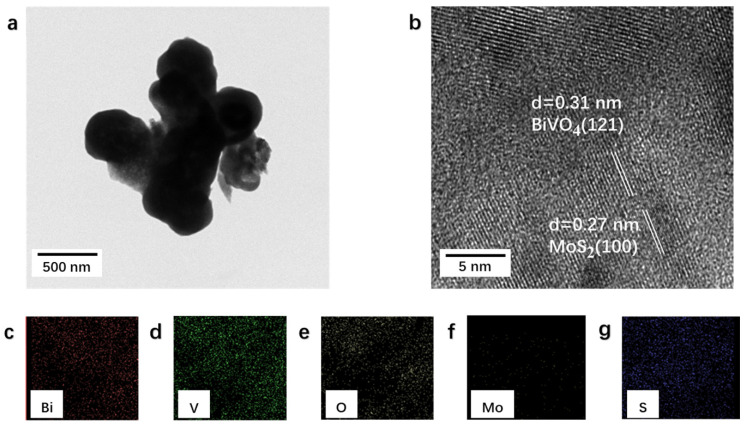
(**a**,**b**) HRTEM image and (**c**–**g**) SEM-EDS elemental mapping of MB5.

**Figure 5 nanomaterials-13-01522-f005:**
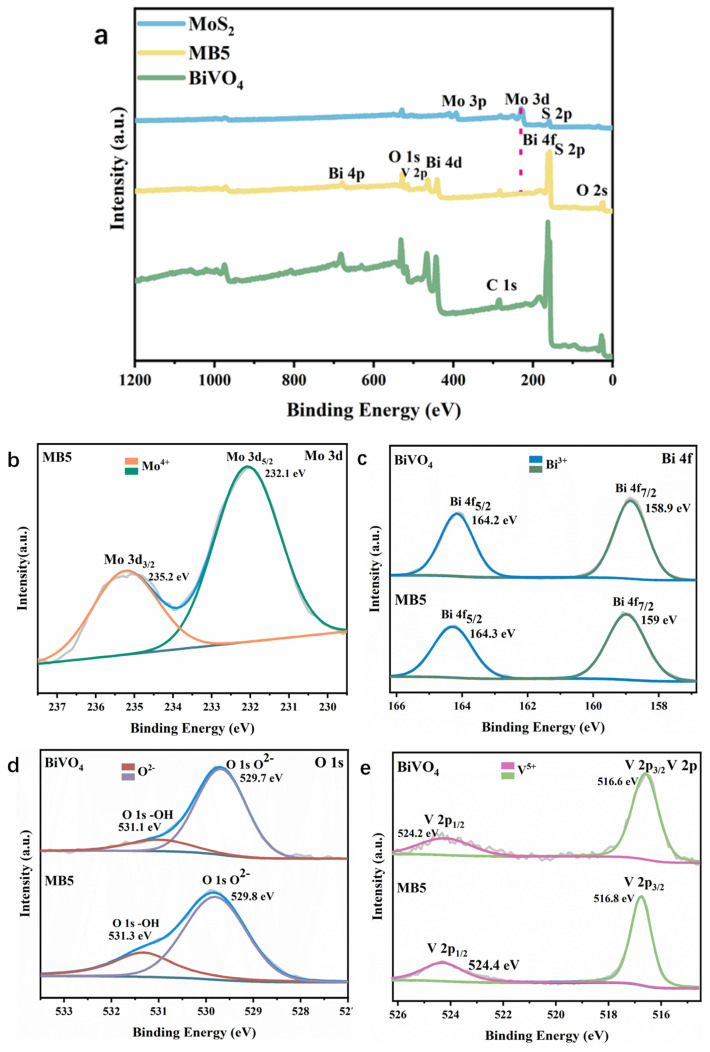
XPS spectra: (**a**) overall spectra of MoS_2_, BiVO_4_ and MB5, (**b**) Mo 3d spectra, (**c**) Bi 4f spectra, (**d**) O 1s spectra, and (**e**) V 2p spectra of BiVO_4_ and MB5.

**Figure 6 nanomaterials-13-01522-f006:**
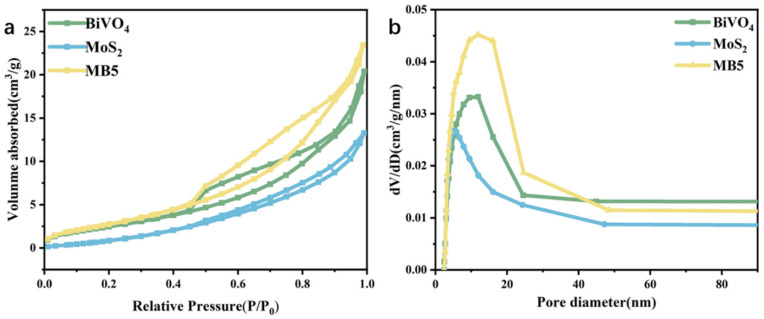
(**a**) N_2_ adsorption-desorption curves and (**b**) pore size distribution of MoS_2_, BiVO_4_ and MB5.

**Figure 7 nanomaterials-13-01522-f007:**
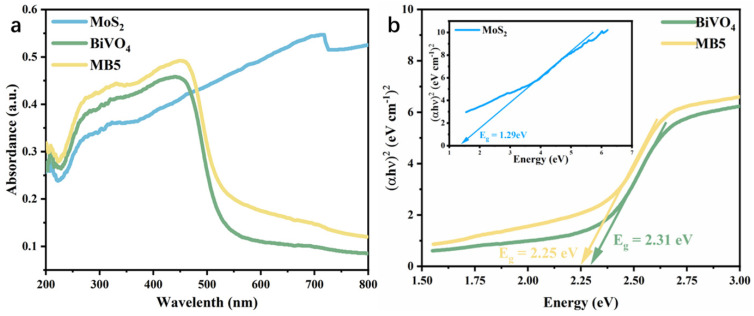
(**a**) UV–vis absorption spectra and (**b**) the plot of (αhν)^2^ vs. energy hν and band gap energy of t of MoS_2_, BiVO_4_ and MB5.

**Figure 8 nanomaterials-13-01522-f008:**
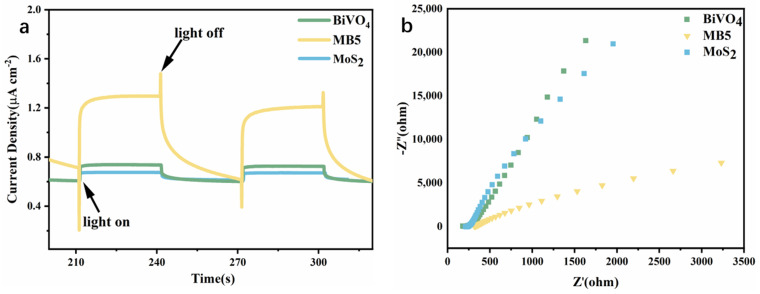
(**a**) Photocurrent response (1 V vs. Ag/AgCl) under visible light irradiation (λ > 420 nm) and (**b**) EIS spectra of MoS_2_, BiVO_4_ and MB5.

**Figure 9 nanomaterials-13-01522-f009:**
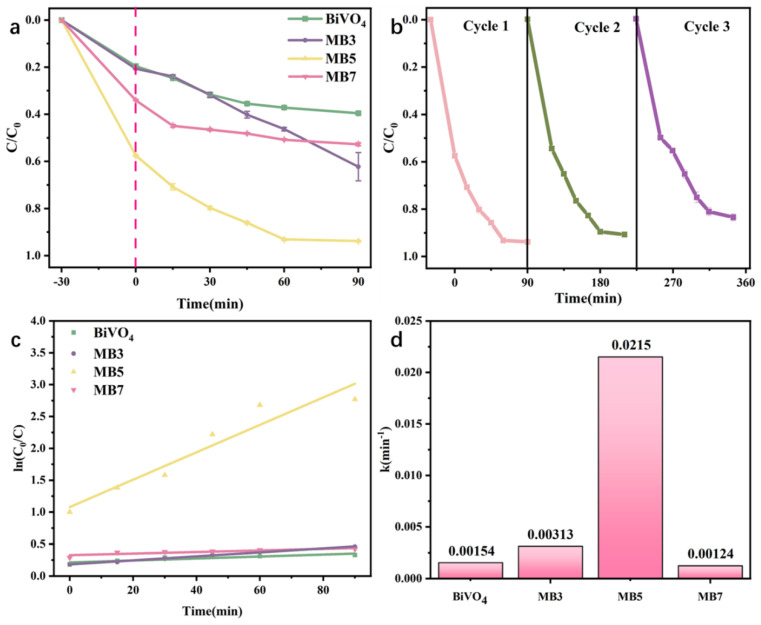
(**a**) Degradation efficiency of TC in visible light, (**b**) Recycle degradation of TC, (**c**) quasi first-order kinetics of degradation of TC, and (**d**) the kinetic constants k of degradation of TC (MB5 = 50 mg, TC = 100 mL, 5 mg L^−1^, time = 90 min).

**Figure 10 nanomaterials-13-01522-f010:**
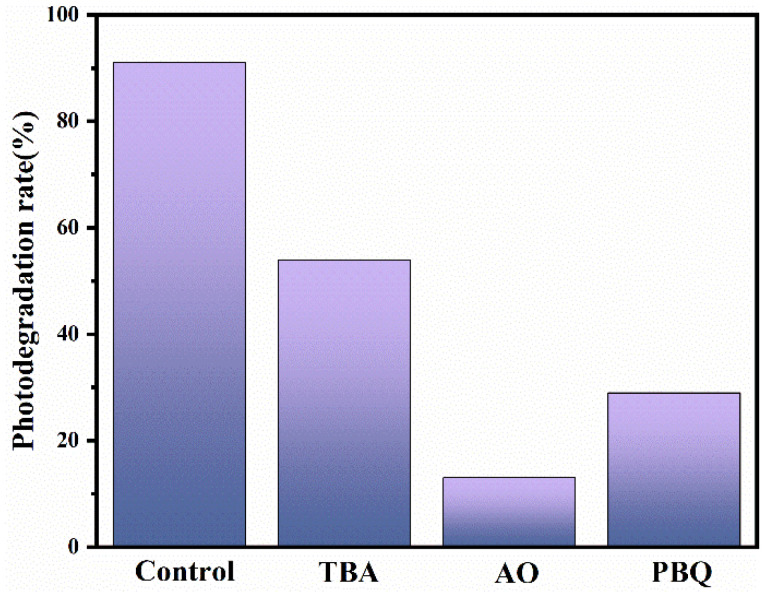
Active species capture tests of MB5 in photocatalytic degradation (TBA = 1.8 mL, AO = 0.3 mmol, PBQ = 0.3 mmol).

**Figure 11 nanomaterials-13-01522-f011:**
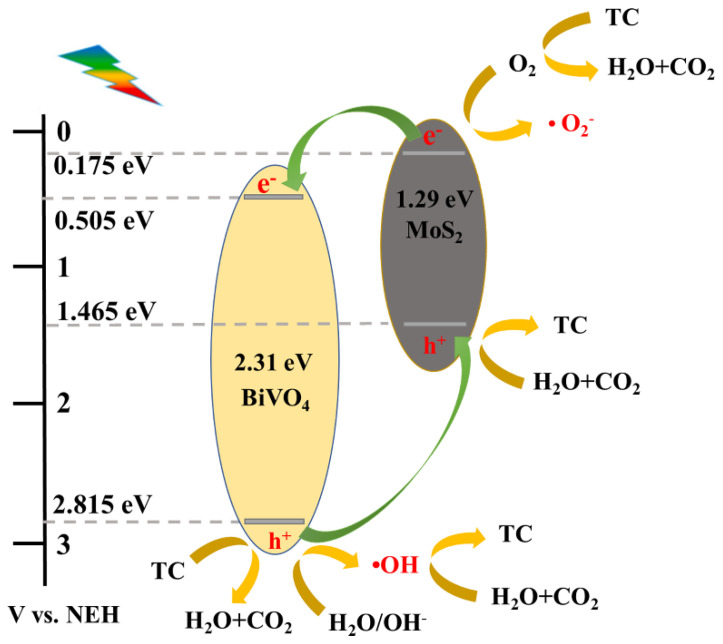
Mechanism of TC degradation by MoS_2_/BiVO_4_ heterojunction photocatalyst.

## Data Availability

The data that support the findings of this study are available from the corresponding author upon reasonable request.

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
