# Peer review of "Microwave-Assisted Synthesis of MoS2/BiVO4 Heterojunction for Photocatalytic Degradation of Tetracycline Hydrochloride"

_nanomaterials, 2023, doi:10.3390/nano13091522_

Round 1

Reviewer 1 Report

In this work the authors prepared MoS2/BiVO4 for the application of photocatalytic degradation of TCH antibiotic. The work presented by authors is great and the prepared material was characterized thoroughly. In this regard I have only minor some comments before recommending this manuscript for publication.

Comments:

1.      It would better for reader if authors summarized the composition obtained by SEM-EDS in the table.

2.      In the BET section, the authors have to explain the reason behind the improvement of the surface area of the MB% catalyst.

3.      In the photocatalytic activity, the catalyst concentration was not identified!!

4.      Please add more data in the photocatalytic application part. The application result is very limited. Since it is the core of this article, the authors have to add more results such as catalyst performance for TCH degradation under different pH, catalyst dosage study and TCH concentration study…

Moderate editing of English language is required

Author Response

  1. In SEM-EDS analysis, the peak positions of Bi, Mo, and S are all around 2.2 keV, and the low proportion of Mo and S elements make it difficult to distinguish between these three elements. The SEM-EDS total spectrum obtained from our test was shown in the following, which is consistent with the relevant literature (Journal of Alloys and Compounds. 2022. 929, 167252.).

Supplemental figure SEM-EDS analysis of MB5. (The picture can be found in the attachment.)

  1. Under the influence of microwave energy, MoS2 nanospheres are broken, resulting in a smaller size, which in turn increases the specific surface area. In the synthesis process of MB5, BiVO4 and small-sized MoS2 are subjected to a microwave reaction, and the increased specific surface area of MoS2 leads to an overall increase in the specific surface area of MB5 (ACS Appl. Mater. Interfaces 2017, 9, 34, 29203–29212). Based on the suggestion, we added this reason to the paper. For your convenience, we hereby attach the added paragraph of the BET as follows:

(Page 10, Line 264)

The findings indicated that MB5 had the largest specific surface area. This was because in the synthesis process of MB5, MoS2 were broken down into smaller-sized MoS2 by microwave energy. Through the microwave reaction, the increased surface area of the MoS2 led to an overall increase in the specific surface area of MB5[51].

  1. The catalyst concentration used in this experiment is 50 mg. For your convenience, we hereby attach the added paragraph of the catalyst concentration as follows:

(Page 3, Line 139)

Specifically, 50 mg of photocatalyst was added to 100 ml of TC solution (5 mg L-1) was stirred in dark for 30 min to achieve the adsorption-desorption equilibrium. And then turned on the light illuminated 90 min for photocatalytic reaction. The absorption at 356 nm was used to analyze the TC concentration[35].

  1. Thank you for raising this question. The focus of this work is to optimize and prepare photocatalysts with excellent photocatalytic performance. We are currently conducting an independent research on the application of the photocatalysts in another separate paper, which will provide a more comprehensive and systematic investigation.

Reviewer 2 Report

1. The language quality must be improved. There are too many mistakes. For example, in Figure 2 b, it should be wavenumber, rather than wavenuber.

2. Abstract is not well written.

3. The photocurrent and EIS of MoS2 should be given.

4. All the figure captions need more details.

The language must be edited with professional help.

Author Response

  1. We have revised all the English language and spelling errors in the manuscript, and issues with other graphics, such as the spelling error in Fig. 2b, have been corrected. For your convenience, we hereby attach the added paragraph of Fig. 2b as follows:

(Page 5, Line 199)

Figure 2. (b) FT-IR spectra of MoS2, BiVO4 and MB5. (The picture can be found in the attachment.)

2. Thank you for your feedback. We have revised the abstract of the article accordingly. For your convenience, we hereby attach the added paragraph of the abstract as follows:

Compared with traditional hydrothermal synthesis, microwave-assisted synthesis has the advantages of faster and more energy efficient. In this work, MoS2/BiVO4 heterojunction photocatalyst was synthesized by microwave-assisted hydrothermal method within 30 minutes. The morphology, structure and chemical composition were characterized by X-ray diffraction (XRD), Raman, X-ray photoelectron spectroscopy (XPS), scanning electron microscope (SEM), and high-resolution transmission electron microscopy (HRTEM). The results of characterizations demonstrated that the synthesized MoS2/BiVO4 heterojunction was a spherical structure with dimensions in the nanorange. In addition, the photocatalytic activity of the samples was investigated by degrading tetracycline hydrochloride (TC) under visible light irradiation. Results indicated that MoS2/BiVO4 heterojunction significantly improved the photocatalytic performance compared with BiVO4 and MoS2, in which the degradation rate of TC (5 mgL-1) by compound which the mass ratio of MoS2 and BiVO4 was 5 wt% (MB5) was 93.7% in 90 min, which was 2.36 times of BiVO4. The active species capture experiments indicated that •OH, •O2- and h+ active species play a major role in the degradation of TC. The degradation mechanism and pathway of the photocatalysts were proposed through the analysis of the band structure and element valence state. Therefore, microwave technology provided a quick and efficient way to prepare MoS2/BiVO4 heterojunction photocatalytic efficiently.

3. Thank you for your suggestion. We have added the curve of MoS2 to the photocurrent and EIS images, and attached the revised images below. Based on your suggestion, we added I-T and EIS curves of MoS2 in the paper. For your convenience, we hereby attach the added paragraph of the EIS as follows:

(Page 11, Line 302)

To further investigate the charge separation of MoS2/BiVO4 heterojunction, the photocurrent response and electrochemical impedance spectroscopy (EIS) of MoS2, BiVO4 and MB5 were measured (Fig. 8). Fig. 8a showed that, MB5 displayed the highest photocurrent response (1.3 μA cm-2) under visible light (λ>420 nm) illumination, which was 3.25 times that of MoS2 and 1.85 times that of BiVO4. A higher photocurrent response of the MB5 suggests the increased charges separation efficiency, which increased lifetime of the photogenerated charge and thus improved the photocatalytic rate[56,57]. In the EIS curve, when compared with MoS2 and BiVO4, it was evident that MB5 had the smallest arc radius (Fig. 8b), which was consistent with the results the photocurrent density test. The minimum arc radius implies that MB5 has the lowest resistance and highest charge transfer efficiency. The aforementioned results suggested that the separation efficiency and transport performance of photogenerated carriers in heterojunction were hugely enhanced.

Figure 8. (a) Photocurrent response (1 V vs. Ag/AgCl) under visible light irradiation (λ>420 nm) and (b) EIS spectra of MoS2, BiVO4 and MB5. (The picture can be found in the attachment.)

4. In order to facilitate readers' understanding of the research work in this paper, we have supplemented the figure captions of all the graphs.

Reviewer 3 Report

Manuscript entitled “Microwave-assisted synthesis of MoS2/BiVO4 heterojunction for photocatalytic degradation of tetracycline” is written with quite good English, however some mistakes still appeared in the text, i.e.:

-          Line 48: “[…] sensitivity visible light response […].

-          Line 58: “[…] to construct an La-BiVO4/CN step-scheme heterojunction photocatalyst […]” – should be “a” instead of “an”.

-          Line 192: “[…] the peak at 416 cm-1 was produced by the […]” – I think peaks cannot be “produced”.

-          Figure 2a: x-axis name contains a typo.

-          Line 217: “Through HRTEM further detect the microstructure of MoS2/BiVO4 heterostructures” – the verb is missing.

Moreover, below please find my other questions and comments:

-          In total, Authors showed in the manuscript 5 samples: MoS2, BiVO4 and three MoS2/BiVO4 heterojunctions differed by MoS2 content. However, even after such low amount of samples, Authors do not show the complete set of results, just “picking” some results. It is not possible to fully compare the materials properties with shredded set of data.

-          Why Authors did not try to synthesis the MoS2 with the use of microwave method?

-          “3”, “5” and “7” represents the percentage of MoS2 or BiVO4 mass percentage? Chemicals weights shown in the text suggest the MoS2, however the sentence: (“”3” meant the mass ratio between MoS2 and BiVO4 was 3 wt%”) rather suggests BiVO4.

-          There is no information about FT-IR in chapter 2.4.

-          In Introduction, chapter 2.5 and further in photocatalytic results, Authors show that ·OH, h+ and ·O2- are “mainly responsible for photocatalytic TC degradation”. However, no other scavengers were tested, so how Authors could tell, those are “mainly” responsible?

-          XRD, FT-IR and Raman results are surprisingly weak for pure MoS2, almost as this compound was amorphous. So, how it is possible, that for MB7 XRD MoS2 peaks are several times more intense than for pure material?

-          Figure 4c-g mapping pictures are almost illegible.

-          What is the meaning of different N2 adsorption-desorption curves’ shapes? (Figure 6a)

-          What is the cause of decreasing and then once again increasing BiVO4 crystallite sizes?

-          Did Authors check physicochemical composition of heterojunction MB5 sample after photocatalytic cycles?

-          What is the path of TC photodegradation?

-          Please add reference link and access date to “Environmental Performance Index” (Reference no. 1).

Concluding, I suggest major revision of presented article, with possibility of acceptance after some corrections.

All comments regarding English Language are included in the main Comments and Suggestions part.

Author Response

  1. The grammatically incorrect sentences in the article have been modified, as shown below. (Page 2, Line 48)

BiVO4 as an n-type semiconductor, is considered to be a promising visible-light-driven photocatalyst with a low band gap (Eg=2.4 eV), sensitive visible light response, strong chemical stability, and nontoxicity[12].

  1. The grammatically incorrect sentences in the article have been modified, as shown below. (Page 2, Line 58)

Jin et al.[17] used a self-assembly method to construct a La-BiVO4/CN step-scheme heterojunction photocatalyst and applied the photocatalyst to degrade TC

  1. The grammatically incorrect sentences in the article have been modified, as shown below. (Page 5, Line 194)

For BiVO4, the peak at 416 cm−1 corresponds to the chemical stretching of Bi-O.

  1. Maybe you are referring to Figure 2b? Corrected the spelling error in Figure 2a.

  1. The grammatically incorrect sentences in the article have been modified, as shown below. (Page 6, Line 220)

Further investigation of the microstructure of MoS2/BiVO4 heterostructures was conducted through HRTEM (Fig. 4).

  1. According to our photocatalytic experiment results, MB5 exhibits the best photocatalytic performance. Therefore, we selected MB5 for other characterizations.

  1. Thank you for raising the issue. We have made multiple attempts to synthesize MoS2 using microwave-assisted methods, but XRD results show that we were not successful in this synthesis. The temperature limit of our microwave instrument is 50 ℃ below the boiling point of the solvent used. When water was used as a solvent and heated to 150 ℃, MoS2 was not successfully synthesized. We later switched to ethylene glycol as a solvent and increased the reaction temperature to 200℃, but still failed to synthesize MoS2. Below is the XRD spectrum of our attempted synthesis of MoS2 using the microwave-assisted method.

Supplemental figure XRD patterns of MOS2 prepared by microwave irradiation. (The picture can be found in the attachment.)

  1. “3”, “5” and “7” represents the percentage of MoS2 mass percentage. And, we have revised the sentence to: "3" meant the mass ratio between MoS2 and BiVO4 was 3 wt%.

  1. Added information related to FT-IR. Using FT-IR spectrometer (IRAffinity-1s, Shimadzu, Japan) with the scan range from 2000 to 400 cm−1 and the resolution of 4 cm-1 to test the chemical bond type and structure of materials.

(Page 5, Line 194)

10:According to some top journals (Chemosphere. 2022. 289, 133158. Chemosphere 2021, 263, 128279. Sep. Purif. Technol. 2022, 299.), the most likely active species for MoS2/BiVO4 materials is ·OH, h+, and ·O2-. Therefore, the use of TBA, PBQ and AO as quenching agents in experiments is the most important capture method for the active species corresponding to this material. Therefore, we did not consider the presence of other active species.

11: According to our research, under microwave conditions, the impact of microwave energy further improves the integrity of MoS2 crystals. In addition, the proportion of MoS2 added in MB7 is larger than that in MB3 and MB5, which increases the diffraction peak intensity of MoS2 in MB7.

  1. In the SEM-EDS analysis, the peak positions of Bi, Mo, and S were all around 2.2 keV, and the proportion of Mo and S elements was relatively low, making it difficult to distinguish these three elements. It is also possible that the area we selected for testing was not ideal, resulting in a blurry SEM-EDS result.

  1. According to the International Union of Pure and Applied Chemistry (IUPAC) standards, different types of adsorption isotherms represent different internal pore structures. The different shapes of N2 adsorption-desorption isotherms reflect information about the sample's pore structure, pore size, and distribution, and can therefore be used to characterize material pore characteristics. Materials with "H1-type" curves have relatively small-radius straight pore channels or capillaries. This curve shape indicates that the material has mainly micropores and/or narrow straight pores. Materials with "H2-type" curves have increasing pore sizes and larger quantities, with pores showing a certain degree of connectivity. This curve shape indicates that the material has mainly mesopores and/or wide straight pores. Materials with "H3-type" curves indicate that the material has larger pore sizes and a wider range of pore distributions. This curve shape indicates that the material has mainly large pore-size pores and/or pore clusters. Attached is the reference (Journal of the American Chemical Society, 1938, 60(2): 309-319. Academic Press, London, 1982. Advances in Catalysis, 1948, 1: 55-98).

  1. The size of the prepared BiVO4/MoS2 composite material is related to the amount of MoS2 added. When the amount of MoS2 added is relatively small, it provides a substrate for the growth of BiVO4, which facilitates the nucleation and crystal growth of BiVO4 and reduces the size of the composite material. However, with the continuous addition of MoS2, too much MoS2 may cause the BiVO4 particles to re-aggregate into larger particles, thereby increasing the size of the composite material.

  1. Thank you for raising this question. The focus of our previous work was to find highly efficient photocatalysts with excellent photocatalytic performance. We are independently conducting another paper for the follow-up work of using photocatalysts to degrade pollutants, which aims to conduct a more systematic and comprehensive study.

  1. The pathway of TC degradation by photocatalysis is the subject of our next work, and we will conduct a more systematic exploration in future studies. In this article, we proposed the mechanism of photocatalytic degradation of tetracycline hydrochloride by referring to literature on the degradation of tetracycline hydrochloride using MoS2/BiVO4 photocatalysts.

  1. Added the access link for Environmental Performance Index 2022.

https://epi.yale.edu/epi-results/2022/overall-ranking

(Page 15, Line 397)

Reviewer 4 Report

The authors conduct a research about microwaved assisted synthesis BiVO4/MoS2. The results is interesting, and the content is nicely present. However, there are some suggestions and comments that is need to be considered.

1. In the introduction the authoors said that "the photocatalytic 49 performance of BiVO4 has limited by the high photogenerated carriers recombination rate 50 and weak surface adsorption ability[13]". Therefore, it is better to emphesize the heterostructure formation of BiVO4/MoS2 regarding solving thus issues.

2. Why the XRD peaks intensity became more crystalline when the amount of MoS2 was increased?

3. The Raman peaks of MoS2 is not clear, would you please to make an inset figure for MoS2 Raman vibrations.

4. Its seems that the adsorption equilibrium had not yet reach after 30 minutes, it`s better to wait until the adsorption equilibrium is reached to ensure the photocatalytic reaction process.

5. There is a typo in Fig. 10.

Author Response

  1. Thank you for your question. We further elaborated on the contribution of heterojunction formation to photocatalytic performance in the conclusion.

(Page 15, Line 377)

Compared with BiVO4, the heterojunction structure formed by MoS2/BiVO4 has stronger adsorption capacity, greater visible light utilization, smaller band gap and higher separation of photogenerated carriers, and these advantages synergistically lead to the enhanced photocatalytic activity of the MoS2/BiVO4.

  1. According to our research, under microwave conditions, the impact of microwave energy further improves the integrity of MoS2 crystals. Moreover, the addition of MoS2 to the composite material increases the diffraction peak intensity.

  1.  I have made an inset figure for MoS2 Raman vibrations.

Figure 2. (a) Raman spectrum of MoS2, BiVO4, MB3, MB5 and MB7. (The picture can be found in the attachment.)

  1. Thank you for your question. We attempted to conduct dark adsorption on MB5 for one hour before starting the photocatalytic experiment. The results showed that the absorbance of TC remained almost unchanged during the dark adsorption period of 30 to 60 minutes. According to literature, an adsorption-desorption equilibrium can be reached after 30 minutes. Attached below is the result of the photocatalytic experiment after one-hour dark adsorption.

Supplemental figure Degradation efficiency of TC in visible light after 60 minutes of dark adsorption (MB5=50 mg, TC=100 ml, 5mg L-1).  (The picture can be found in the attachment.)

  1. We have revised all the English language and spelling errors in the manuscript, and issues with other graphics, such as corrected typing errors in Figure 10.

Figure 10. Active species capture tests of MB5 in photocatalytic degradation.  (The picture can be found in the attachment.)

Round 2

Reviewer 1 Report

the authors improved the manuscript significantly and addressed most my comments. I can recommend the article for the publication

Minor editing of English language required

Reviewer 3 Report

Authors answered all my questions and doubts as well as corrected the manuscript. In this regard, I suggest acceptance of presented manuscript in its present form.